**Data availability statement:** All primary datasets were obtained from the National Health

# Is there a competitive advantage to using multivariate statistical or machine learning methods over the Bross formula in the hdPS framework for bias and variance estimation?

**Mohammad Ehsanul Karim**[1,2]*, Yang Lei[3]

**1** School of Population and Public Health, University of British Columbia, Vancouver, British Columbia, Canada, **2** Centre for Advancing Health Outcomes, St. Paul's Hospital, Vancouver, British Columbia, Canada, **3** Department of Statistics, University of British Columbia, Vancouver, British Columbia, Canada

* ehsan.karim@ubc.ca

## Abstract

**Purpose**: We aim to evaluate various proxy selection methods within the context of high-dimensional propensity score (hdPS) analysis. This study aimed to systematically evaluate and compare the performance of traditional statistical methods and machine learning approaches within the hdPS framework, focusing on key metrics such as bias, standard error (SE), and coverage, under various exposure and outcome prevalence scenarios. **Methods:** We conducted a plasmode simulation study using data from the National Health and Nutrition Examination Survey (NHANES) cycles from 2013 to 2018. We compared methods including the kitchen sink model, Bross-based hdPS, Hybrid hdPS, LASSO, Elastic Net, Random Forest, XGBoost, and Genetic Algorithm (GA). The performance of each inverse probability weighted method was assessed based on bias, MSE, coverage probability, and SE estimation across three epidemiological scenarios: frequent exposure and outcome, rare exposure and frequent outcome, and frequent exposure and rare outcome. **Results:** XGBoost consistently demonstrated strong performance in terms of MSE and coverage, making it effective for scenarios prioritizing precision. However, it exhibited higher bias, particularly in rare exposure scenarios, suggesting it is less suited when minimizing bias is critical. In contrast, GA showed significant limitations, with consistently high bias and MSE, making it the least reliable method. Bross-based hdPS, and Hybrid hdPS methods provided a balanced approach, with low bias and moderate MSE, though coverage varied depending on the scenario. Rare outcome scenarios generally resulted in lower MSE and better precision, while rare exposure scenarios were associated with higher bias and MSE. Notably, traditional statistical approaches such as forward selection and backward elimination performed comparably to more sophisticated machine learning methods in terms of bias and coverage, suggesting that these simpler approaches may be viable alternatives due to their computational efficiency. **Conclusion:** The results highlight the importance of selecting hdPS methods

and Nutrition Examination Survey (NHANES), which can be accessed through the NHANES website:

https://www.cdc.gov/nchs/nhanes/index.htm. The analysis datasets and source code used in this study are available at

https://github.com/ehsanx/hdPS-proxy-select-codes, with a stable and citable version archived on Zenodo (DOI: 10.5281/zenodo.15201233). Additionally, all simulation results can be interactively explored via a Shiny web application at

https://ehsanx.shinyapps.io/hdPS-Alternatives/.

**Funding:** This work was supported by MEK's Natural Sciences and Engineering Research Council of Canada (NSERC) Discovery Grant (RGPIN-2018-05044) and Discovery Launch Supplement (DGECR-2018-00235). The funders had no role in study design, data collection and analysis, decision to publish, or preparation of the manuscript.

**Competing interests:** MEK is currently supported by grants from the Canadian Institutes of Health Research (CIHR) and MS Canada. MEK has previously received consulting fees from Biogen Inc. for consulting work unrelated to the current study and was also previously supported by a Scholar Award from the Michael Smith Foundation for Health Research. This does not alter our adherence to PLOS policies on sharing data and materials.

based on the specific characteristics of the data, such as exposure and outcome prevalence. While advanced machine learning methods such as XGBoost can enhance precision, simpler methods such as forward selection or backward elimination may offer similar performance in terms of bias and coverage with fewer computational demands. Tailoring the choice of method to the epidemiological scenario is essential for optimizing the balance between bias reduction and precision.

## Background

### High-dimensional Propensity Score (hdPS) algorithm

Propensity scores are a crucial tool in causal inference for addressing confounding. A propensity score is the conditional probability of receiving a particular treatment or exposure, given a set of observed covariates. Traditional propensity score models rely on "investigator-specified" or manually selected covariates based on domain knowledge, prior literature, or theoretical frameworks.

In epidemiology, proxy variables are commonly used as substitutes for confounders that are difficult or impossible to measure directly, such as socioeconomic status, lifestyle factors, or health behaviors [1]. These variables are not directly identified as confounders but serve as surrogates for unmeasured or poorly measured confounders. The hdPS is a pharmacoepidemiological method designed to reduce confounding bias in large healthcare databases [2]. HdPS automatically ranks a wide array of proxy variables from healthcare records—such as diagnosis codes, medications, and procedures—using the Bross formula [3,4]. The Bross formula ranks these variables based on their marginal associations with both exposure and outcome. These selected proxy variables serve as surrogates for unmeasured or poorly measured confounders, helping reduce bias in treatment effect estimates. The hdPS algorithm further refines the selection by prioritizing variables based on their prevalence and potential for confounding, defined by their association with both exposure and outcome [2].

Importantly, hdPS has become a foundational method in modern epidemiologic research that uses healthcare databases, offering a scalable and reproducible approach to confounding control when traditional investigator-specified propensity score models are limited by unmeasured, mismeasured, or omitted variables. Unlike conventional PS methods that rely solely on expert knowledge and a fixed set of covariates, hdPS empirically identifies and prioritizes proxy variables from high-dimensional data, allowing it to adjust for latent confounding structures more effectively. This makes hdPS particularly valuable for improving causal inference in observational healthcare studies, including those evaluating drug safety, treatment effectiveness, and health system interventions [5,6].

### Multivariate machine learning extensions

Although the Bross formula performs well in certain contexts, it has limitations in capturing complex interactions, nonlinearities, and higher-order associations between variables, especially in high-dimensional settings where it does not account for the multivariate structure of other covariates [5,7]. To address these model-specification-related limitations, multivariate machine learning methods such as LASSO, Elastic Net, and Random Forests have been applied within the hdPS framework. These methods are better suited for high-dimensional data, where they can more effectively handle complex relationships and improve the selection of proxy variables, thus enhancing the precision of treatment effect estimates [7–9]. Several studies have proposed hybrid approaches that combine machine learning and hdPS

to enhance variable selection and bias reduction while maintaining interpretability [6,7,9]. Simulation studies and empirical research have shown that these machine learning-based methods, or hybrid approaches combining the Bross formula with machine learning, can reduce confounding more effectively and increase efficiency compared to the Bross formula alone in certain settings [7–9]. These efforts reflect a growing interest in refining hdPS with modern statistical learning tools to optimize proxy confounder adjustment in realistic, data-rich settings.

### Assessing the simulation performance

Other than bias, previous studies have primarily focused on Mean Squared Error (MSE) as a key metric for evaluating the performance of hdPS and its machine learning extensions [7,9–12]. However, in high-dimensional settings with singly robust methods, such as hdPS and machine learning approaches such as LASSO, MSE may not always be the most reliable measure. MSE combines both bias and variance into a single metric, which makes interpretation challenging when variance estimation is unstable—a common issue in these methods. In contrast, coverage, which measures the proportion of confidence intervals that capture the true treatment effect, provides a more direct and meaningful assessment of a model's reliability. In realistic observational studies, where model misspecification is often inevitable, coverage—along with related metrics such as bias-eliminated coverage and relative error in standard error (SE; which compares model-based SE with empirical SE)—can reveal whether confidence intervals or SEs are too narrow (underestimating uncertainty) or too wide (overestimating uncertainty). This insight is crucial in determining whether the model delivers valid estimates despite misspecification. Even a model with poor MSE but good coverage may still be valuable, as it produces realistic confidence intervals. By shifting the focus to coverage, rather than relying solely on MSE, we can achieve a more comprehensive understanding of method performance, especially in cases where unstable variance estimation might distort conclusions drawn from MSE alone. To extend prior research, our study systematically compares multiple machine learning and hybrid hdPS approaches using plasmode simulations grounded in real-world data structure. Importantly, we expand beyond commonly reported metrics such as bias and MSE by incorporating underutilized diagnostics—namely, empirical coverage and relative error in standard error—to provide a more nuanced understanding of estimator performance.

**Aim**: This research aims to systematically evaluate and compare various proxy selection methods within the hdPS framework for inverse probability weighted estimators, using a diverse range of simulation performance metrics, including bias, MSE, and coverage. The study focuses on assessing how these alternative statistical and machine learning methods perform in selecting proxy variables for confounding adjustment, compared to the traditional Bross formula.

## Methods

### Data and simulation

**Motivating example**: We revisited the association between obesity and the risk of diabetes using data from three cycles of the National Health and Nutrition Examination Survey (NHANES) covering the years 2013-2014, 2015-2016, and 2017-2018 [5]. To identify relevant investigator-specified covariates, we constructed a causal diagram based on literature [13–16] and established causal inference principles [17]. The covariates included in our analysis were carefully selected and categorized into demographic, behavioral, health history, access-related,

and laboratory variables. While most of these variables were binary or categorical, the Laboratory variables were continuous. The secondary, de-identified data used in this study were accessed from the NHANES database on 1/5/2024-31/8/2024. The authors did not have access to any information that could identify individual participants during or after data collection. NHANES data are fully anonymized and publicly available, ensuring that individual privacy is protected.

**Plasmode simulation**: To rigorously assess the performance of the methods under consideration, we employed a plasmode simulation framework, which is particularly well-suited for reflecting real-world data structures and complexities [18]. This approach was inspired by the analytic dataset derived from NHANES and involved resampling from the observed covariates and exposure information (i.e., obesity) without altering them. By mirroring key aspects of an actual epidemiological study, this simulation framework offers a substantial advantage over traditional Monte Carlo simulations, which often rely on hypothetical assumptions.

**Simulation scenarios under consideration**: Our plasmode simulation was conducted over 500 iterations. For the base simulation scenario, we set the prevalence of exposure (obesity) and the event rate (diabetes) at 30%, with a true odds ratio (OR) parameter of 1, corresponding to a risk difference (RD) of 0. Each simulated dataset had a sample size of 3,000 participants. The description of other scenarios under consideration is provided in Table 1.

**True data generating mechanism used in plasmode simulation**: The primary goal of this plasmode simulation study is to evaluate various variable selection methods under realistic conditions. To achieve this, we formulated the outcome data based on a specific model specification that incorporates both exposure and covariates, including investigator-specified and proxy variables. The model specification consists of three key components (See Appendices §A and B for further details):

1. *Investigator-specified covariates (demographic, behaviour and health history / access variables)*: We retained the original investigator-specified covariates, which were either binary or categorical, reflecting how real-world studies typically operate. These covariates were not included in the pool of proxy variables or subjected to any variable selection or ranking processes. Instead, they were always included in the analysis as a separate set of variables based on prior knowledge and causal reasoning. This ensured that these known confounders were appropriately accounted for, independent of the proxy variable selection framework.

2. *Investigator-specified covariates (transformation of laboratory variables)*: In real-world studies, it is common for analysts to lack precise knowledge of the true model specification. To simulate this uncertainty, we transformed the continuous laboratory variables using complex functions such as logarithmic, exponential, square root, polynomial transformations, and interactions. This reflects the challenges analysts face in correctly specifying models when dealing with continuous data.

**Table 1. Overview of plasmode simulation scenarios reflecting varying exposure and outcome prevalences based on National Health and Nutrition Examination Survey (NHANES) Data Cycles (2013–2018).**

| Plasmode simulation scenario | Exposure Prevalence | Outcome Prevalence | True Odds ratio | Sample Size |
|---|---|---|---|---|
| (i) Frequent exposure and outcome (base) | 30% | 30% | 1 | 3,000 |
| (ii) Rare exposure and frequent outcome | 5% | 30% | 1 | 3,000 |
| (iii) Frequent exposure and rare outcome | 30% | 5% | 1 | 3,000 |

3. *Inclusion of proxy variables*: Real-world studies often deal with unmeasured confounding, which researchers attempt to mitigate by adding proxy variables. However, when a large number of proxies are added, some may act as noise variables, contribute little or nothing to the analysis. To simulate this, we selected only those binary proxy covariates (referred to as recurrence covariates in hdPS terminology) that had a relative risk (RR) of less than 0.8 or greater than 1.2 concerning the outcome. Proxy variables were defined based on their association with the outcome because unmeasured confounders must influence the outcome to induce confounding. By prioritizing variables with strong outcome associations, we aimed to ensure that the proxies reflect at least part of the confounding structure. While not explicitly enforced, many of the selected proxy variables are expected to have some degree of association with the exposure.

Out of 142 proxy covariates, 94 met this criterion and were summed in calculating a simple comorbidity burden measure [19]. This score represents the individual's overall comorbidity burden, reflecting the cumulative presence of conditions. The remaining 48 covariates were excluded from this calculation and considered noise. This comorbidity burden measure (one variable) was then incorporated into our model specification for generating the outcome
data.

**Performance measures**: From this simulation, we derived several performance metrics to evaluate the effectiveness of the methods under consideration: (1) bias, (2) average model-based SE (the average of estimated SEs obtained from a model over repeated samples), (3) empirical SE (the standard deviation of estimated treatment effects across repeated samples), (4) MSE, (5) coverage probability of 95% confidence intervals, (6) bias-corrected coverage, and (7) Zip plot [20,21].

\textcolor{black}{Bias-corrected coverage provides a more accurate reflection of model reliability when bias is present. By adjusting for bias, we isolate the coverage performance attributable to variance estimation alone, allowing clearer comparisons of uncertainty calibration. This correction is crucial for distinguishing methods that produce narrow but biased intervals from those that offer valid inference under realistic misspecification.

While MSE is a commonly used metric, its interpretation becomes less reliable when variance estimates are unstable, as observed with singly robust machine learning models. In such cases, MSE may mask the trade-off between high variance and low bias (or vice versa), especially when confidence intervals are miscalibrated. Our inclusion of coverage and standard error diagnostics addresses this limitation by providing complementary insights into inference accuracy.

## Inverse probability weighted estimators under consideration

The comparison between the data generation process and the analysis process reveals two key differences: (i) The data generation used transformed laboratory variables, whereas the analysis was conducted using only the original laboratory variables. (ii) The data generation employed a simple sum of selected proxy variables (sum of 94 proxy covariates), while the analysis included all proxy variables (142 binary proxies), with 48 of these acting as noise variables. These differences help us assess how the proxy variable selection methods handle model misspecification and the presence of noise variables.

We estimated propensity scores using the following models separately. Propensity scores were then used to construct weights for each individual, defined as the inverse of the

probability of their observed exposure status. The primary causal estimand of interest in this study is the average treatment effect (ATE), which represents the difference in expected outcomes between treated and untreated individuals if all had been exposed to either treatment condition.

The selected machine learning models were chosen for their demonstrated ability to handle high-dimensional data and perform feature selection effectively. For example, XGBoost and Random Forest are tree-based ensemble methods known for capturing non-linear relationships and interactions. Genetic Algorithm (GA) offers a stochastic, optimization-based approach to variable selection, allowing exploration of complex covariate spaces. These models align with the goals of hdPS by enabling automated, multivariate proxy variable selection that may capture complex dependencies missed by univariate selection strategies like the Bross formula. Incorporating such models expands the flexibility and potential robustness of hdPS analyses in high-dimensional data settings.

1. **Kitchen sink model**: The kitchen sink model serves as a comprehensive baseline that includes all available covariates—both investigator-specified and proxy variables—without any variable selection. It is used to benchmark the performance of variable selection methods against a model that assumes complete adjustment [7].

2. **hdPS using Bross formula**: The Bross formula is a statistical method used to calculate the bias introduced by not adjusting for a covariate [4]. In hdPS analysis, this formula was originally applied to each proxy variable to measure and rank the potential bias if the covariate were not adjusted for. In our analysis, the 100 proxies with the highest bias rankings are selected for further modeling [2,3]. This threshold of 100 proxy variables has been used in the literature to balance model complexity and overfitting risk, while ensuring that a sufficient number of proxies are included to capture unmeasured confounding [22]. Our choice reflects a practical trade-off between dimensionality and computational feasibility.

3. **Least Absolute Shrinkage and Selection Operator (LASSO)**: LASSO is a variable selection technique that limits the number of variables by adding a penalty term to the regression model. Cross-validation (CV) is used in LASSO to identify variables with non-zero coefficients in the best model by optimizing the penalty value [7–9].

4. **Hybrid of hdPS and LASSO**: Instead of relying solely on LASSO for variable selection, a hybrid approach combines the Bross formula and LASSO. First, proxy variables are selected using the hdPS algorithm (e.g., the top 100), and then LASSO is applied to further refine the selection [7,9].

5. **Elastic Net**: Elastic Net is an extension of LASSO that includes an additional penalty term to handle multicollinearity by grouping correlated features and selecting the most representative ones [7].

6. **Random Forest**: The Random Forest algorithm is an ensemble learning method that constructs multiple decision trees to perform classification [23]. It calculates the importance of each proxy variable based on the decrease in impurity or Gini importance, providing a ranking of the proxies. The top 100 variables from this ranking are manually selected for further modeling [8].

7. **XGBoost**: XGBoost is a gradient boosting algorithm used to optimize machine learning models [24]. It builds decision trees that make splits based on maximum impurity reduction, and it assigns an importance score to each proxy variable by calculating the mean decrease in impurity [25].

8. **Stepwise**: Stepwise selection is a commonly used variable selection technique that builds or reduces models by adding or removing predictors based on statistical

criteria—in our case, adjusted R-squared. We implemented two versions: (a) Forward selection (FS) begins with a minimal model, typically including only investigator-selected covariates. It then adds one proxy variable at a time—the one that most improves model performance at each step. This process continues until no additional variable meaningfully improves the fit. (b) Backward elimination (BE) takes the opposite approach. It starts with a full model containing all investigator-selected and proxy variables and removes the least informative variable at each step, based on its contribution to model fit. This continues until removing more variables no longer improves or begins to harm the model. These stepwise approaches are transparent, making them accessible alternatives for confounding adjustment in high-dimensional settings.

9. **Genetic algorithm**: Genetic algorithm is an evolutionary algorithm inspired by the theory of natural selection [26]. It operates by evolving offspring from a population of the fittest individuals over several generations, evaluating and selecting the best combination of features or variables that maximize prediction accuracy.

## Ethics approval and consent to participate

The analysis conducted on secondary and de-identified data is exempt from research ethics approval requirements. Ethics for this study was covered by item 7.10.3 in University of British Columbia's Policy no. 89: Research and Other Studies Involving Human Subjects 19 and Article 2.2 in of the Tri-Council Policy Statement- Ethical Conduct for Research Involving Humans (TCPS2).

## Results

The results for each method under the different scenarios are summarized below. See Figs 1 and 2 for an overview of the performance in terms of bias and coverage, respectively. All simulation results can be reviewed interactively through a Shiny web application available at https://ehsanx.shinyapps.io/hdPS-Alternatives/, providing a convenient platform for exploring the performance of each method across various scenarios. Analysis data and codes are provided in https://github.com/ehsanx/hdPS-proxy-select-codes, with a stable, citable release archived at Zenodo (DOI: 10.5281/zenodo.15201233 and URL: https://doi.org/10.5281/zenodo.15201233).

(i) **Frequent Exposure and Outcome (base) scenario**:

1. *Bias*: Bross-based hdPS exhibited the smallest bias (-0.0001), and the kitchen sink model (0.0002) was the second. Genetic algorithm shows the highest bias (0.0287), indicating a substantial deviation from the true effect. Among the other methods, Hybrid hdPS (0.0016), and Elastic Net (0.0036) demonstrated low bias. XGBoost (0.0074) had slightly higher bias than Random Forest (0.0034).

2. *Coverage*: The coverage for most methods was high, with Hybrid hdPS, Forward Selection, Backward Elimination, LASSO, and Elastic Net achieving values around 98%, indicating well-calibrated confidence intervals. However, Genetic algorithm had noticeably lower coverage (83.8%), indicating that its confidence intervals might be too narrow or biased, potentially missing the true effect. After applying bias elimination (as there were substantial bias associated with this method), Genetic algorithm's coverage improved to 96%.

## Comparison of Bias Across Scenarios.

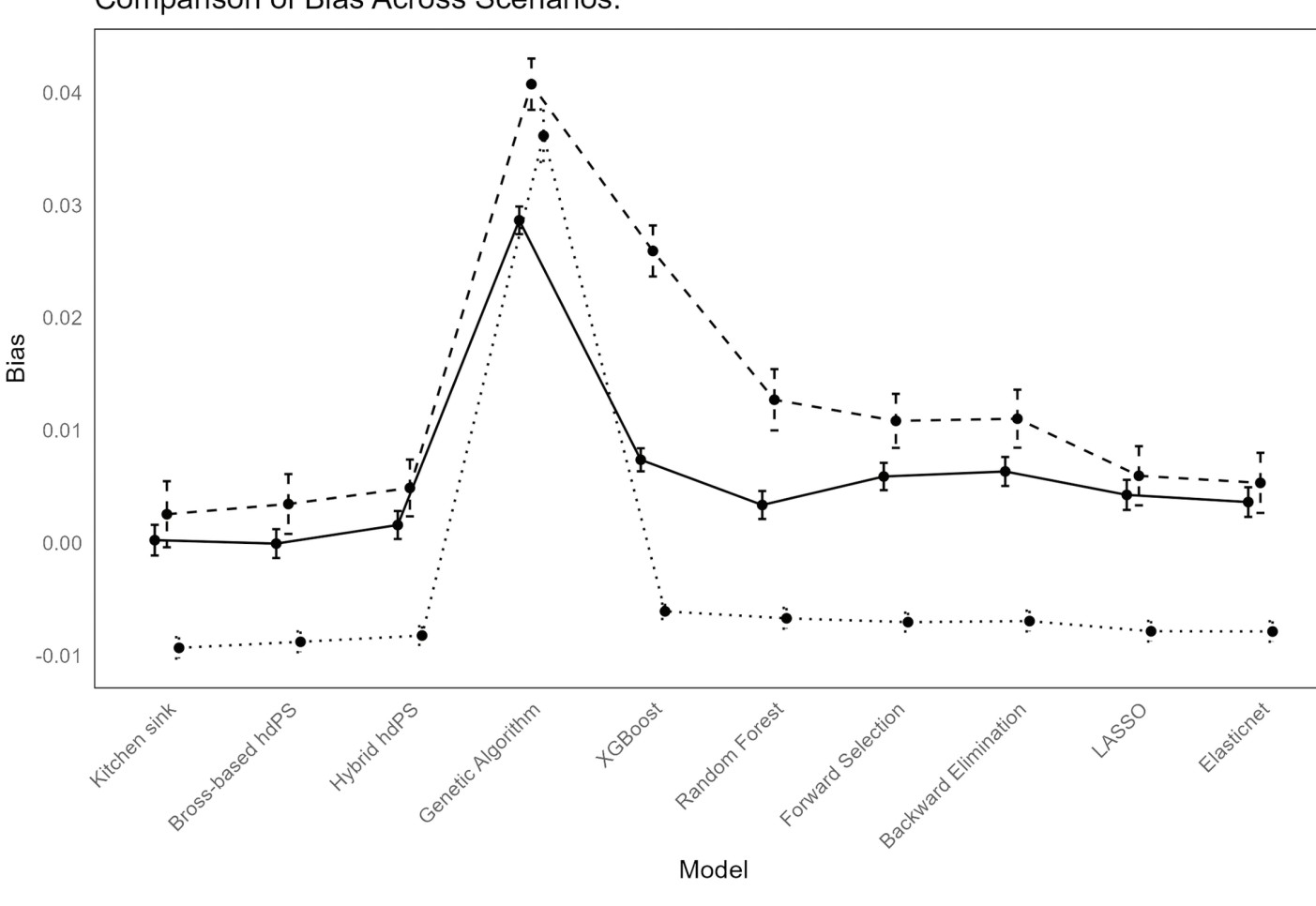

**Fig 1.** Comparison of Bias Across Different Methods in hdPS Analysis.

3. *MSE*: XGBoost achieved the lowest MSE (0.0006). Genetic algorithm maintained the highest MSE (0.0016), reflecting its higher bias and variability. The kitchen sink model (0.0009), Bross-based hdPS (0.0008), Hybrid hdPS (0.0008), and Elastic Net (0.0009) all had relatively similar and moderate MSE values.

4. *SE*: XGBoost exhibited the lowest Empirical SE (0.0229), indicating high precision in its estimates. The kitchen sink model had the highest Empirical SE (0.0305), suggesting greater variability. Other methods, including Genetic algorithm (0.0274), Hybrid hdPS (0.0278), and Bross-based hdPS (0.0287), showed moderate variability. LASSO (0.0299) and Elastic Net (0.0294) had slightly higher variability. The Model-based SE followed a similar pattern, with XGBoost (0.0268) showing the lowest variability and the kitchen sink model (0.0333) the highest, indicating less precision in its estimates. When comparing relative error in SE estimation, XGBoost performed the worst. See Appendix §C for further details.

## Comparison of Coverage Across Scenarios.

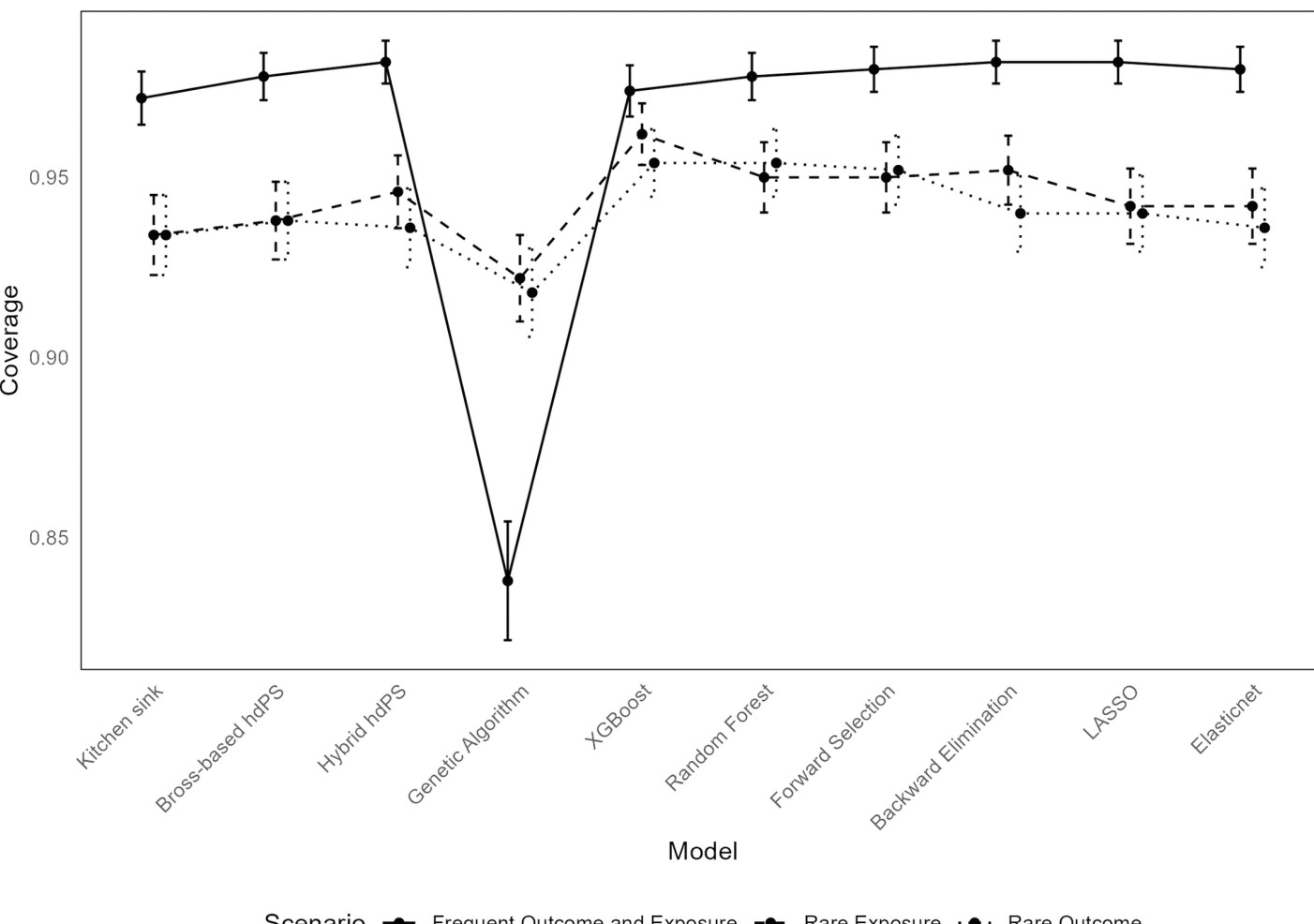

**Fig 2.** Comparison of Coverage Probability Across Different Methods in hdPS Analysis.

**(ii) Rare Exposure and Frequent Outcome**:

1. *Bias*: The kitchen sink model showed a relatively low bias (0.0025), while Genetic algorithm continued to exhibit the highest bias (0.0408), indicating a significant deviation from the true effect. XGBoost had a bias of 0.0259, which, while still higher than other methods, but was lower than Genetic algorithm. Other methods, such as Bross-based hdPS (0.0035), Hybrid hdPS (0.0049), and Elastic Net (0.0053), demonstrated moderate bias. Random Forest (0.0127) and Forward Selection (0.0108) had slightly higher bias but remained within an acceptable range.

2. *Coverage*: Coverage levels remained high for most methods, with XGBoost achieving the highest coverage at 96.2%, indicating well-calibrated confidence intervals despite its higher bias. The Genetic algorithm method had lower coverage (92.2%), suggesting that its confidence intervals might be narrower, potentially missing the true effect. Other methods such as RF, Forward Selection, Backward Elimination, and Hybrid hdPS maintained coverage values around 94-95%, suggesting adequate interval calibration.

Bias-eliminated coverage for Genetic algorithm improved to 94.2%, though it remained slightly lower than that of other methods (e.g., forward selection).

3. *MSE*: Forward selection demonstrated the lowest MSE (0.0030), and then Hybrid hdPS and XGBoost. The Genetic algorithm method had a higher MSE (0.0043), reflecting its substantial bias and variability. The kitchen sink model also had an MSE of 0.0043, similar to Genetic algorithm, while other methods such as Bross-based hdPS (0.0035), RF (0.0039), and Elastic Net (0.0036) exhibited moderate MSE values, indicating reasonable accuracy.

4. *SE*: The lowest Empirical SE was observed with XGBoost (0.0507) and Genetic algorithm (0.0510), reflecting high precision despite their higher bias. The kitchen sink model had the highest Empirical SE (0.0656), indicating greater variability. Hybrid hdPS (0.0564), Bross-based hdPS (0.0595), and RF (0.0609) showed moderate variability. Forward Selection (0.0537) and Backward Elimination (0.0576) had lower variability compared to the kitchen sink model. In terms of Model-based SE, XGBoost (0.0531) and Genetic algorithm (0.0533) continued to show low variability, while the kitchen sink model had the highest Model-based SE (0.0623), indicating less precision in its estimates. When comparing relative error in SE estimation, XGBoost and kitchen sink model performed the worst (in other direction).

**(iii) Frequent Exposure and Rare Outcome**:

1. *Bias*: In this scenario, the kitchen sink model exhibited a moderate negative bias (-0.0093), similar to the Bross-based hdPS method (-0.0088). Genetic algorithm showed a significantly higher bias (0.0362), indicating a substantial deviation from the true effect. Among other methods, XGBoost demonstrated the lowest bias (-0.0061), while methods such as Hybrid hdPS (-0.0082), Forward Selection (-0.0070), and Backward Elimination (-0.0070) had slightly higher but still moderate biases. Elastic Net and LASSO both had biases of -0.0079, reflecting slightly larger deviations compared to XGBoost but still within acceptable limits.

2. *Coverage*: Most methods achieved good coverage, with XGBoost, RF, and Forward Selection each achieving a coverage rate of 95.4%, indicating well-calibrated confidence intervals. The Genetic algorithm method, however, had slightly lower coverage (91.8%), indicating that its confidence intervals might be narrower, potentially excluding the true effect. Bross-based hdPS and the kitchen sink model had slightly lower coverage values of 93.8% and 93.4%, respectively. After accounting for bias, the bias-eliminated coverage for most methods, except Genetic algorithm, remained high, with values ranging from 98.4% to 99.0%, indicating that most methods effectively adjusted for bias in their coverage estimates. Genetic algorithm's bias-eliminated coverage was lower at 93.4%, reflecting its higher inherent bias.

3. *MSE*: XGBoost exhibited the lowest MSE (0.0003). Genetic algorithm had the highest MSE (0.0040), reflecting its substantial bias and variability. The kitchen sink model (0.0005), Bross-based hdPS (0.0005), and other methods such as Hybrid hdPS (0.0004) and Elastic Net (0.0005) all had relatively similar MSE values, indicating moderate accuracy.

4. *SE*: The lowest Empirical SE was observed with XGBoost (0.0152), reflecting high precision in its estimates. The Genetic algorithm method exhibited the highest Empirical SE (0.0523), indicating greater variability and less precision. Methods such as Hybrid hdPS (0.0184), Forward Selection (0.0187), and Elastic Net (0.0203) showed moderate variability, while Bross-based hdPS (0.0206) and the kitchen sink model (0.0212) had

slightly higher variability. In terms of Model-based SE, XGBoost (0.0179) again showed the lowest variability, consistent with its low Empirical SE, indicating that it provided the most stable estimates. The kitchen sink model had a slightly higher Model-based SE (0.0219), indicating less precision in its estimates. When comparing relative error in SE estimation, XGBoost performed the worst.

## Real-world analysis

**Summary results**: The dataset comprises 7,585 individuals. Among these, the prevalence of the exposure is 48.8%, while the prevalence of the outcome is 23.7%.

See Fig 3 for the results from analyzing the NHANES (2013-2018) dataset. The methods are arranged according to the number of selected proxy variables. Among all variable selection algorithms, Forward Selection and LASSO yield the highest ORs, with values of 1.56 and 1.55, respectively. The ORs for the remaining methods cluster around 1.50 to 1.54. Additionally, with the exception of a few methods such as Kitchen Sink and Genetic Algorithm, a general pattern emerges in which methods selecting a smaller number of proxy variables tend to produce higher ORs. For RD, Forward Selection again shows the highest value at 0.082, followed closely by LASSO (0.080). Most other methods yield RD values between 0.075 and 0.079, reflecting a similar pattern to that observed for the ORs. This consistency suggests that the relationship between variable selection method and effect size persists across different estimands.

Table 2 presents a pairwise comparison of the number of proxy features shared between different variable selection methods used in the analysis. Each cell in the table indicates the

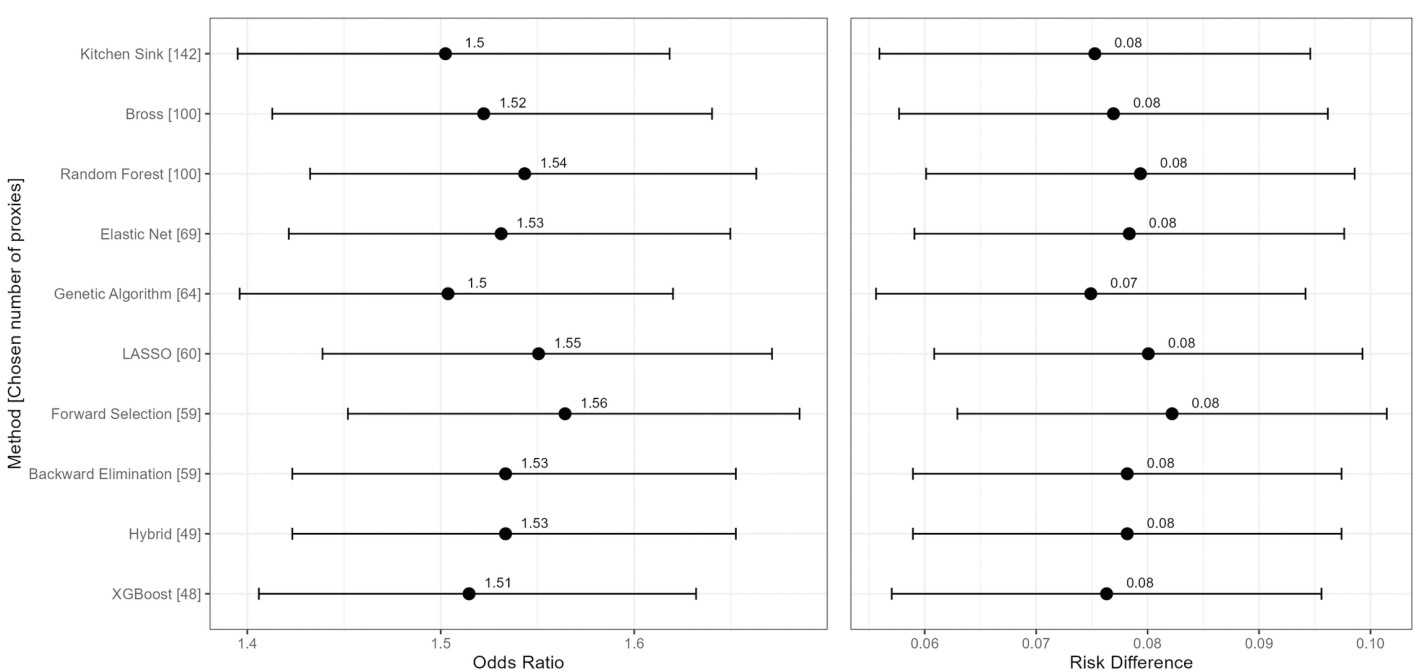

**Fig 3.** Comparison of Risk Differences (RD) and Odds Ratios (OR) with 95% confidence intervals for different methods used to evaluate the association between obesity and diabetes risk. The analysis is based on data from the National Health and Nutrition Examination Survey (NHANES) for the years 2013–2018. Methods are arranged by the number of variables used in the models.

**Table 2. Comparison of variable overlap of selected proxies across different methods used to evaluate the association between obesity and diabetes from the National Health and Nutrition Examination Survey (NHANES) for the years 2013–2018. Diagonal entries show the total number of proxies selected by each method, and off-diagonal entries represent the count of shared variables between method pairs. Most methods share a moderate number of proxies (typically 50-60 percent of the smaller set), indicating partial agreement in variable selection. Higher overlap is observed between closely related methods (e.g., LASSO and Elastic Net, or Hybrid with Bross/LASSO), while methods like XGBoost and Genetic Algorithm show lower overlap with others, reflecting divergent selection behavior in high-dimensional settings.**

| | KS | Bross | Hybrid | LASSO | EN | RF | XGB | FS | BE | GA |
|---|---|---|---|---|---|---|---|---|---|---|
| **Kitchen sink (KS)** | 142 | | | | | | | | | |
| **Bross formula** | 100 | 100 | | | | | | | | |
| **Hybrid (Bross and LASSO)** | 49 | 49 | 49 | | | | | | | |
| **LASSO** | 60 | 47 | 47 | 60 | | | | | | |
| **Elastic Net (EN)** | 69 | 54 | 48 | 60 | 69 | | | | | |
| **Random Forest (RF)** | 100 | 71 | 38 | 46 | 52 | 100 | | | | |
| **XGBoost (XGB)** | 48 | 38 | 24 | 28 | 30 | 37 | 48 | | | |
| **Forward selection (FS)** | 59 | 45 | 41 | 51 | 54 | 45 | 25 | 59 | | |
| **Backward elimination (BE)** | 59 | 45 | 41 | 51 | 54 | 45 | 25 | 59 | 59 | |
| **Genetic algorithm (GA)** | 64 | 44 | 28 | 36 | 40 | 49 | 25 | 35 | 35 | 64 |

count of common proxy variables selected by the method in the corresponding row and column. The diagonal cells, where the row and column methods are the same, represent the total number of proxy variables selected exclusively by each method. The first column displays the number of proxy variables shared between each method and the kitchen sink (KS) model. Given that the KS model includes all proxy variables, the values in the first column are identical to those in the diagonal cells for each row, which also presents the total number of proxy variables selected by the method in the corresponding row.

Table 3 presents a comparison of the count and percentage of common proxy variables selected by different methods in comparison with the Bross formula hdPS. The first column shows the total number of proxy variables selected by each method, while the second column lists the number of common features selected by both the respective method and the Bross

**Table 3. Comparison of the count and percentage of proxy variables selected by each methods in common with that by the Bross formula-based high-dimensional propensity score to evaluate the association between obesity and diabetes from the National Health and Nutrition Examination Survey (NHANES) for the years 2013–2018. The Hybrid method (Bross + LASSO) shows perfect agreement with the Bross-based hdPS by design. Other methods demonstrate overlap rates ranging from 0.69 to 0.79, indicating moderate consistency in variable selection. XGBoost shows the highest overlap (79 percent) among the non-hybrid methods, while the Genetic Algorithm shows the lowest (69 percent), reflecting greater divergence in selected proxies. These overlap patterns highlight methodological differences in how each approach prioritizes covariates in high-dimensional settings.**

| Method | Total Count | Common Count | Rate in common |
|---|---|---|---|
| Kitchen sink | 142 | – | – |
| Bross formula | 100 | – | – |
| Hybrid (Bross and LASSO) | 49 | 49 | 1.00 |
| LASSO | 60 | 47 | 0.78 |
| Elastic Net | 69 | 54 | 0.78 |
| Random Forest | 100 | 71 | 0.71 |
| XGBoost | 48 | 38 | 0.79 |
| Forward selection | 59 | 45 | 0.76 |
| Backward elimination | 59 | 45 | 0.76 |
| Genetic algorithm | 64 | 44 | 0.69 |

formula. The third column reports the percentage of common features out of the total number of features selected by each method.

We observe that, aside from the Bross formula hdPS and Random Forest models, where the number of proxies was manually set to 100, and the kitchen sink (KS) model, which includes all proxies by design, the number of proxy variables selected by other models ranges between 49 and 69. As expected, the hybrid method combining Bross and LASSO hdPS selects exactly 49 proxy variables, all of which are selected by both methods, resulting in a common feature rate of 1.00. For other models, the common feature percentage is generally clustered around 74%, with XGBoost showing the highest common percentage at 79%, while GA displays the lowest common percentage at 69%.

**Computing time**: Fig 4 presents the computing time for each method. All methods, aside from RF and GA, exhibit relatively fast computing times. RF and GA are generally much slower.

## Discussion

### Summary of the simulation findings

**Comparison of methods:** Across the three scenarios, XGBoost consistently achieved the lowest MSE and high coverage, making it one of the most reliable methods in terms of precision. However, it

**Fig 4.** Computing time for the real-world analysis for each algorithm under consideration. The analysis is based on data from the National Health and Nutrition Examination Survey (NHANES) for the years 2013–2018.

consistently exhibited some degree of bias, particularly when compared to methods such as the kitchen sink model, Bross-based hdPS, and Hybrid hdPS, which often showed lower bias in scenarios with frequent outcomes. In contrast, GA displayed the highest bias and MSE, along with lower coverage and greater variability, making it the least reliable method for accurate effect estimation. Methods such as Bross-based hdPS, Hybrid hdPS, and Elastic Net performed moderately well across all scenarios, balancing bias, coverage, and MSE. However, these methods did not outperform XGBoost in terms of overall precision, especially with respect to MSE, though they often resulted in lower bias. The kitchen sink model performed comparably to Bross-based hdPS in terms of bias and coverage but lagged in SE estimation and MSE.

While XGBoost consistently yielded the lowest MSE across scenarios—indicating high overall accuracy—this gain came at the cost of increased bias, particularly in rare exposure settings. This pattern suggests that while XGBoost is highly efficient in minimizing prediction error, it may over-prioritize variance reduction at the expense of bias, potentially due to aggressive regularization or suboptimal feature weighting in the context of non-linear, sparse exposure structures. In causal inference applications, such trade-offs warrant caution, as models with high precision but biased estimates can result in systematically misleading conclusions regarding treatment effects.

GA exhibited consistently high bias and MSE, indicating substantial limitations in handling the dimensionality and noise characteristics inherent in the hdPS framework. One plausible explanation is that GA's stochastic search process lacks the structure to efficiently navigate large covariate spaces with many weakly informative or correlated variables. Unlike regularized regression or tree-based methods, GA does not incorporate intrinsic mechanisms for managing multicollinearity or prioritizing marginally predictive features, which may lead to unstable or suboptimal variable subsets and subsequently poor confounding adjustment. Its computational intensity further compounds these limitations, making it less practical for high-dimensional epidemiologic analyses.

**Comparison of scenarios:** In scenarios with rare exposure, higher bias was observed, particularly for methods such as GA and XGBoost, whereas frequent outcomes generally led to lower bias across most methods. Overcoverage was more common in the scenario with frequent exposure and outcome, with several methods producing confidence intervals that were too wide, suggesting an overestimation of uncertainty. The other scenarios exhibited more balanced or slightly under-coverage. Scenarios with frequent exposure also displayed higher relative error in SE

estimation, making it more difficult to precisely estimate effects. In contrast, rare exposure scenarios were associated with higher MSE, reflecting the challenge of estimating effects when the exposure is uncommon. Rare outcome scenarios exhibited the lowest MSE across most methods (except GA), indicating that these scenarios provided a better balance between bias and precision for most methods.

These findings have practical implications for selecting variable selection strategies in epidemiological research. For studies focused on rare exposures (e.g., occupational or genetic risk factors), methods that minimize bias—such as Bross-based hdPS, Hybrid hdPS, or forward/backward stepwise selection—may be more suitable, as they provide more stable effect estimates under sparse data conditions. In contrast, studies targeting common conditions or aiming to improve precision, such as evaluations of widely used treatments or lifestyle exposures, may benefit from machine learning method such as XGBoost, which consistently demonstrate lower MSE. However, trade-offs between bias and variance should be carefully considered in relation to the study's primary inference goal.

## Contextualizing the literature

Previous studies have shown that LASSO performs well within the hdPS framework, particularly in terms of bias reduction and MSE [7,9], with similar results for Elastic Net [7]. Our findings largely support these conclusions, as both methods demonstrated moderate bias and MSE across the scenarios. However, their performance was inconsistent in some cases, with higher bias observed in rare exposure scenarios, as noted in previous studies [7,9]. MSE also increased in rare exposure scenarios. The Hybrid hdPS method, combining Bross-based hdPS and LASSO, showed promising results, especially in terms of precision (i.e., lower variance), suggesting it may serve as a suitable alternative to traditional hdPS methods. Random Forest also performed similarly in our study, yielding relatively low bias, which is consistent with previous findings [7]. However, none of the earlier works emphasized coverage or examined methods such as GA, XGBoost, forward selection, or backward elimination, which were evaluated here.

## Data analysis findings

The real-world dataset, with frequent exposure (48.8%) and a moderate outcome rate (23.7%), produced ORs of 1.56 and 1.55 for Forward Selection and LASSO, respectively, while other methods clustered around OR values between 1.50 and 1.54. A general trend emerged, showing that methods selecting fewer proxy variables tended to yield higher ORs, with exceptions such as the Kitchen Sink and Genetic Algorithm, which selected more proxies but produced lower ORs. In terms of risk difference (RD), estimates were relatively stable across methods, with Forward Selection again yielding the highest RD (0.082), followed by LASSO (0.080), while others ranged from 0.075 to 0.079. Regarding variable overlap, most methods shared around 74% of their proxy variables with the Bross formula hdPS, with XGBoost showing the highest common rate (79%) and GA the lowest (69%). Computing time analysis showed that, aside from RF and GA, most methods had relatively fast computing times, with RF being significantly slower.

In applied research, particularly with large-scale electronic health records or claims data, computational efficiency is a critical consideration. While tree-based or stochastic optimization methods may offer performance gains, their longer runtime may be impractical in settings requiring rapid analyses or model updates. Simpler approaches such as stepwise selection or penalized regression offer viable alternatives when speed, transparency, or reproducibility are prioritized.

## Strengths

Previous studies on singly robust methods have mainly focused on hdPS performance using MSE as the primary evaluation metric [7,9–12]. Our study extends this body of work by incorporating a broader range of performance metrics, allowing researchers to compare results in terms of both bias and variance estimation. This comprehensive comparison of statistical and machine learning methods across various scenarios has not been conducted before. For instance, we found that while XGBoost consistently demonstrated strong performance in terms of MSE and coverage, it did not always have the lowest bias. Additionally, we observed that traditional variable selection methods such as forward selection and backward elimination performed similarly to more sophisticated methods such as LASSO and Random Forest across scenarios, both in terms of bias and coverage.

Our study also employed a complex plasmode simulation framework, closely replicating real-world data conditions [18]. This framework not only accounted for model

misspecification, where the true relationships between covariates and outcomes were unknown, but also introduced noise variables, addressing the challenge of dealing with irrelevant covariates in high-dimensional settings. By applying transformations to continuous variables and adding proxy variables that contributed minimally to the outcome, we rigorously evaluated how well each method handled both model uncertainty and non-informative variables, further strengthening the real-world relevance of our findings.

## Future direction

Double robust methods have demonstrated strong potential for achieving optimal statistical performance in hdPS analyses [10,27,28]. In addition, single robust methods, such as ensemble learners such as super learners, have shown promise in improving bias, MSE, area under the curve (AUC), and covariate balance in other contexts [11,29,30]. Deep learning methods, particularly supervised architectures, offer significant potential for improving propensity score estimation in high-dimensional settings by capturing complex, non-linear relationships and performing well in scenarios with rare exposures, while maintaining comparable bias and superior variance estimation compared to traditional methods [31]. Despite their theoretical advantages, the complexity and computational demands of these methods, especially in high-dimensional settings, have limited their adoption by practitioners. On the other hand, the simpler singly robust machine learning methods evaluated here have been applied in clinical research [32,33].

Future research should explore the application of double robust methods and super learners in hdPS analyses, particularly for handling rare outcomes and exposures, where the performance of traditional methods may be suboptimal. Additionally, investigating the impact of different hyperparameters on machine learning methods such as XGBoost could optimize their performance in hdPS analysis. Finally, future simulation studies should focus on evaluating coverage and other metrics in epidemiological scenarios such as time-varying exposures, multiple treatment settings, and data with substantial measurement error, which would provide valuable insights into the generalizability of our findings [34].

One avenue for improving bias reduction in machine learning models within hdPS analysis is the incorporation of causal structure or prior knowledge into variable selection. Methods such as targeted maximum likelihood estimation, double machine learning, or feature engineering based on directed acyclic graphs can help align machine learning-based selection with the underlying causal model. Additionally, tuning hyperparameters to favor less aggressive regularization or ensemble weighting may reduce over-smoothing and improve bias performance, particularly in low-prevalence strata.

Hybrid machine learning models—such as combinations of domain-informed methods (e.g., Bross or investigator-specified variables) with data-driven algorithms like LASSO or Random Forest—offer a promising balance between interpretability and predictive accuracy. Their application in health data science could enable more robust causal inference while accommodating real-world data challenges such as missingness, nonlinearity, and high dimensionality. Such models may help mitigate confounding bias without discarding variables of known clinical relevance, improving both validity and stakeholder trust in observational research findings.

## Conclusion

This analysis highlights the importance of carefully selecting appropriate methods for hdPS analysis based on the specific characteristics of the data, particularly the prevalence of exposure and outcome. These findings also emphasize the need to tailor method selection to

the specific epidemiological scenario, ensuring that the chosen method aligns with the study's goals, whether minimizing bias or maximizing precision.

XGBoost consistently demonstrated strong performance in terms of MSE and coverage, making it an effective choice when precision is prioritized. However, it did not achieve the lowest bias, particularly in rare exposure scenarios, indicating that it may be less suited when minimizing bias is the primary objective. In contrast, the GA exhibited significant limitations, with consistently high bias and MSE, making it less reliable for effect estimation.

The kitchen sink, Bross-based hdPS, and Hybrid hdPS methods provided a more balanced approach, offering low bias and moderate MSE, though coverage varied depending on the scenario. The analysis also revealed that rare outcomes were associated with lower MSE and better precision, while rare exposures presented challenges, yielding higher bias and MSE. Interestingly, traditional statistical approaches such as forward selection and backward elimination performed comparably to more sophisticated machine learning methods across many scenarios. This suggests that simpler approaches can still be viable, particularly in terms of bias and coverage, and might be preferred due to their computational efficiency.

In real-world healthcare settings—such as electronic health record-based cohort studies, claims data analysis, or comparative effectiveness research—these results offer concrete guidance on choosing hdPS-compatible variable selection strategies. For instance, institutions conducting routine surveillance or drug safety studies may benefit from hybrid or regularized regression approaches that balance scalability with confounding control. Meanwhile, traditional stepwise methods remain attractive for smaller studies or settings requiring transparent modeling. Overall, aligning method choice with data availability, computational resources, and inferential goals can improve the practical utility of hdPS analyses in healthcare.

## Supporting information

**S1 File**. Supplementary Content.
(PDF)

## Acknowledgment

This research was supported in part through computational resources and services provided by Advanced Research Computing at the University of British Columbia.

## Author contributions

**Conceptualization:** Mohammad Ehsanul Karim.

**Formal analysis:** Yang Lei.

**Funding acquisition:** Mohammad Ehsanul Karim.

**Investigation:** Mohammad Ehsanul Karim.

**Methodology:** Mohammad Ehsanul Karim.

**Resources:** Mohammad Ehsanul Karim.

**Supervision:** Mohammad Ehsanul Karim.

**Validation:** Mohammad Ehsanul Karim.

**Visualization:** Mohammad Ehsanul Karim, Yang Lei.

**Writing – original draft:** Mohammad Ehsanul Karim.

**Writing – review & editing:** Yang Lei.

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
