## [Decision Letter · Decision Letter 0]

24 Mar 2025

PONE-D-25-00468Is There a Competitive Advantage to Using Multivariate Statistical or Machine Learning Methods Over the Bross Formula in the hdPS Framework for Bias and Variance Estimation?PLOS ONE

Dear Dr. Karim,

Thank you for submitting your manuscript to PLOS ONE. After careful consideration, we feel that it has merit but does not fully meet PLOS ONE’s publication criteria as it currently stands. Therefore, we invite you to submit a revised version of the manuscript that addresses the points raised during the review process.

We look forward to receiving your revised manuscript.

Kind regards,

Hossein Ali Adineh, Ph.D

Academic Editor

PLOS ONE

 [This work was supported by MEK’s Natural Sciences and Engineering Research Council of Canada (NSERC) Discovery Grant (PG#: 20R01603) and Discovery Launch Supplement (PG#: 20R12709).]. 

[MEK is currently supported by grants from Canadian Institutes of Health Research and MS Canada. MEK has previously received consulting fees from Biogen Inc. for consulting unrelated to this current work. MEK was also previously supported by the Michael Smith Foundation for Health Research Scholar award.].

5. Thank you for uploading your study's underlying data set. Unfortunately, the repository you have noted in your Data Availability statement does not qualify as an acceptable data repository according to PLOS's standards.

Additional Editor Comments:

Reviewer 1

The manuscript titled "Is There a Competitive Advantage to Using Multivariate Statistical or Machine Learning Methods Over the Bross Formula in the hdPS Framework for Bias and Variance Estimation?" offers a comprehensive evaluation of various proxy selection methods within the high-dimensional propensity score (hdPS) framework. It compares traditional statistical methods with machine learning approaches, and it effectively assesses their performance across various epidemiological scenarios. The use of a plasmode simulation with NHANES data adds significant value to the study's methodology and conclusions.

However, I believe that the manuscript requires major revisions to improve its quality and robustness. The following points outline the areas that need attention:

Introduction and Literature Review: The introduction could be enhanced by providing more context on the critical role of hdPS in modern epidemiology. Specifically, expanding on how hdPS addresses the limitations of traditional propensity score methods and directly impacts causal inference in healthcare data would strengthen the manuscript. The literature review could also benefit from a clearer connection to prior research, particularly focusing on how previous studies have applied machine learning to hdPS and how this study brings new insights to the field.

Methods Section: While the methodology is generally well-detailed, there is a need for further clarification regarding the choice of machine learning models (e.g., Genetic Algorithm, XGBoost) and statistical methods. A better explanation of why these particular models were selected and their relevance to hdPS would make the study's approach more transparent. The "kitchen sink model" is also mentioned briefly but needs further elaboration on its role in the analysis. Furthermore, providing a clearer explanation of the traditional variable selection methods, such as "forward selection" and "backward elimination," would benefit readers who may not be familiar with these techniques.

Results and Discussion: The simulation results are well-presented, but additional discussion on the interpretation of bias and MSE would help contextualize the trade-offs involved in using machine learning methods like XGBoost. While XGBoost shows lower MSE, the increase in bias should be explored in more detail to help readers understand the model's limitations. Similarly, the poor performance of the Genetic Algorithm should be explained with greater depth, especially regarding its struggles with high-dimensional data compared to other methods.

Tables and Figures: The readability of Table 2 and Table 3 could be improved by adding brief summaries or interpretations of the key results in their captions. This would help readers quickly grasp the significance of the findings and how they support the broader argument.

Conclusions: The conclusions are generally strong, but they could be expanded to provide more practical guidance for future research and real-world applications. Specifically, the authors could discuss which methods might be more appropriate for different types of epidemiological studies, such as those focused on rare diseases versus common conditions.

These revisions are essential to enhancing the manuscript’s clarity, depth, and impact. A more robust discussion of the trade-offs between methods, along with clearer explanations of model choices, will help solidify the paper’s contribution to the field.

Recommendation: Given the importance of the topic, I recommend that the authors address the major revisions outlined above. Once these revisions are made, I believe the manuscript could make a valuable contribution to the literature on hdPS and its applications in health data science.

Thank you for considering my review.

Reviewer 2

The manuscript presents a highly relevant comparison of hdPS variable selection methods, including both traditional statistical and machine learning approaches. The findings are useful for epidemiologists and health data scientists.

Key suggestions for revision:

- Clarify the rationale behind selecting 100 proxy variables in hdPS methods.

- Expand the discussion on bias correction for coverage probability.

- Address concerns regarding MSE interpretation when variance estimation is unstable.

- Discuss the practical implications of computational efficiency for real-world applications.

- Improve clarity and conciseness in the Results and Discussion sections.

Reviewers' comments:

Reviewer's Responses to Questions

**Comments to the Author**

1. Is the manuscript technically sound, and do the data support the conclusions?

Reviewer #1: Yes

Reviewer #2: Yes

2. Has the statistical analysis been performed appropriately and rigorously? 

Reviewer #1: Yes

Reviewer #2: Yes

3. Have the authors made all data underlying the findings in their manuscript fully available?

Reviewer #1: Yes

Reviewer #2: Yes

4. Is the manuscript presented in an intelligible fashion and written in standard English?

Reviewer #1: Yes

Reviewer #2: Yes

5. Review Comments to the Author

Reviewer #1: The manuscript presents a highly relevant comparison of hdPS variable selection methods, including both traditional statistical and machine learning approaches. The findings are useful for epidemiologists and health data scientists.

Key suggestions for revision:

- Clarify the rationale behind selecting 100 proxy variables in hdPS methods.

- Expand the discussion on bias correction for coverage probability.

- Address concerns regarding MSE interpretation when variance estimation is unstable.

- Discuss the practical implications of computational efficiency for real-world applications.

- Improve clarity and conciseness in the Results and Discussion sections.

Reviewer #2: The manuscript titled "Is There a Competitive Advantage to Using Multivariate Statistical or Machine Learning Methods Over the Bross Formula in the hdPS Framework for Bias and Variance Estimation?" provides a comprehensive evaluation of various proxy selection methods within the high-dimensional propensity score (hdPS) framework. It compares traditional statistical methods with machine learning approaches, and it effectively assesses their performance in terms of bias, mean squared error (MSE), coverage, and standard error (SE) across different epidemiological scenarios. The study’s use of a plasmode simulation based on NHANES data provides a valuable contribution to the understanding of these methods' efficacy. While the overall analysis is robust, there are several areas where the manuscript could be improved to increase its clarity, depth, and contribution to the literature. Please refer to the comments document as attached for more specifics.

6. PLOS authors have the option to publish the peer review history of their article (what does this mean?). If published, this will include your full peer review and any attached files.

Reviewer #1: No

Reviewer #2: No

---

## [Author Response · Author response to Decision Letter 1]

12 Apr 2025

A PDF is attached that contains the point-by-point response. Also cover letter includes journal requirement related responses.

---

## [Decision Letter · Decision Letter 1]

29 Apr 2025

Is There a Competitive Advantage to Using Multivariate Statistical or Machine Learning Methods Over the Bross Formula in the hdPS Framework for Bias and Variance Estimation?

PONE-D-25-00468R1

Dear Dr. Karim,

We’re pleased to inform you that your manuscript has been judged scientifically suitable for publication and will be formally accepted for publication once it meets all outstanding technical requirements.

Kind regards,

Hossein Ali Adineh, Ph.D

Academic Editor

PLOS ONE

Additional Editor Comments (optional):

Reviewers' comments:

Reviewer's Responses to Questions

**Comments to the Author**

1. If the authors have adequately addressed your comments raised in a previous round of review and you feel that this manuscript is now acceptable for publication, you may indicate that here to bypass the “Comments to the Author” section, enter your conflict of interest statement in the “Confidential to Editor” section, and submit your "Accept" recommendation.

Reviewer #2: All comments have been addressed

2. Is the manuscript technically sound, and do the data support the conclusions?

Reviewer #2: Yes

3. Has the statistical analysis been performed appropriately and rigorously? 

Reviewer #2: Yes

4. Have the authors made all data underlying the findings in their manuscript fully available?

Reviewer #2: Yes

5. Is the manuscript presented in an intelligible fashion and written in standard English?

Reviewer #2: Yes

6. Review Comments to the Author

Reviewer #2: The authors have thoughtfully and thoroughly addressed all the major concerns raised during the previous review. Clear explanations were added for the kitchen sink model, the challenges observed with the Genetic Algorithm (GA), and the role of forward and backward selection methods. The Introduction and Literature Review sections have been strengthened, offering a clearer connection between traditional limitations in propensity score methods, the emergence of high-dimensional propensity score (hdPS) techniques, and the growing role of machine learning approaches in this area.

The Methods section has been clarified, with a well-justified rationale for the choice of machine learning models. In the Results section, the authors provide a more nuanced interpretation of the trade-offs between bias and mean squared error (MSE), and offer a thoughtful explanation for the poorer performance of the GA. Improvements to Table 2 and Table 3, particularly the addition of interpretative summaries in the captions, have enhanced the accessibility and meaning of the results.

The Conclusion and Future Directions sections now provide valuable practical guidance for researchers considering the application of these methods to real-world epidemiological studies, as well as thoughtful suggestions for how machine learning models could be refined for better bias reduction. Overall, the revisions have substantially improved the clarity, depth, and relevance of the manuscript.

I commend the authors for their careful and considered revisions, and I have no further concerns.

7. PLOS authors have the option to publish the peer review history of their article (what does this mean?). If published, this will include your full peer review and any attached files.

Reviewer #2: No

---

## [Editor Report · Acceptance letter]

PONE-D-25-00468R1

PLOS ONE

Dear Dr. Karim,

I'm pleased to inform you that your manuscript has been deemed suitable for publication in PLOS ONE. Congratulations! Your manuscript is now being handed over to our production team.

Kind regards,

on behalf of

Dr. Hossein Ali Adineh

Academic Editor

PLOS ONE